# Understanding Recent Developments in Colistin Resistance: Mechanisms, Clinical Implications, and Future Perspectives

**DOI:** 10.3390/antibiotics14100958

**Published:** 2025-09-24

**Authors:** Shreya Singh, Rajesh Kumar Sahoo, Mahesh Chandra Sahu

**Affiliations:** 1Centre for Biotechnology, Siksha ‘O’ Anusandhan Deemed to Be University, Kalinganagar, Bhubaneswar 751003, Odisha, India; shreya9430064610@gmail.com (S.S.); rajeshkumarsahoo@soa.ac.in (R.K.S.); 2Division of Microbiology, ICMR-Regional Medical Research Centre, Chandrasekharpur, Bhubaneswar 751023, Odisha, India

**Keywords:** colistin resistance, *MCR* genes, antibiotic stewardship, public health, MDR, PmrA

## Abstract

Colistin resistance, driven by chromosomal mutations and the spread of plasmid-mediated *MCR* genes, has emerged as a critical challenge in combating multidrug-resistant Gram-negative bacteria. This resistance compromises the efficacy of colistin, leading to higher treatment failure rates, prolonged hospitalizations, and increased mortality. Recent studies have highlighted key mechanisms, including lipid A modifications, that enable bacteria to evade colistin’s effects. The global spread of *MCR* genes exacerbates the issue, underlining the need for improved diagnostics and rapid detection of resistant strains to prevent adverse patient outcomes. To combat this growing threat, a multifaceted approach is essential, involving enhanced antimicrobial stewardship, stricter infection control measures, and continued research into alternative therapies and diagnostic methods. Collaborative efforts from researchers, healthcare providers, policymakers, and the pharmaceutical industry are crucial to preserving colistin’s effectiveness and mitigating the broader impact on public health.

## 1. Introduction

The rise of multidrug-resistant bacteria has led to greater reliance on colistin, a reserve antibiotic. Increased use has led to development of resistance to colistin itself, which is problematic given that it is one of the last remaining drugs with activity against some multidrug-resistant strains. Antimicrobial resistance (AMR) is among the top global health threats as identified by the WHO [1,2].

Since the pipeline for new antibiotic discoveries is closing and there will not be any soon to treat these “superbugs”, there is a renewed interest in bringing back older drugs that were thought to be toxic for clinical utilization. In particular, the polymyxins (colistin and polymyxin B) should be used as “last resort” antimicrobials [2]. The 77th World Health Assembly recognized the urgent need for evidence to inform public health policy as one of the WHO’s strategic and operational goals to fight drug-resistant bacterial infections between 2025 and 2035 in a resolution on accelerating national and international responses to AMR, which was approved on 30 May 2024 [3]; due to high rates of morbidity and mortality, longer hospital stays (LOS), higher direct medical expenses, and increased societal infectious costs, the resolution urges WHO member states to collaborate with academia, the private sector, and civil society to support and advance basic, applied, and implementation research on antimicrobial stewardship, vaccines, diagnostic tools, treatments, and infection prevention and control. New antibiotics are therefore desperately needed, especially ones that are effective against Gram-negative “superbugs” [3].

One of the major challenges is the acquisition and spread of plasmid-mediated resistance genes, specifically the *MCR-1* to *MCR-10* genes, which are the main causes of the sharp rise in colistin ineffectiveness [4]. These genes encode enzymes capable of modifying the bacterial cell membrane and reducing colistin’s antimicrobial activity. Understanding the mechanisms underlying colistin resistance, along with its clinical implications and potential future directions, is essential for addressing this evolving public health threat.

Due to high rates of morbidity and mortality, longer hospital stays (LOS), higher direct medical expenses, and higher societal infectious costs, AMR can result in significant financial losses [5]. Gram-negative bacteria, such as *Enterobacteriaceae*, *Acinetobacter baumannii*, and *Pseudomonas aeruginosa*, pose a growing threat to international health. On a worldwide scale, it is concerning that resistance to widely used antibiotics, such as beta-lactams, carbapenems, fluoroquinolones, and aminoglycosides, is emerging quickly. In light of the lack of equally effective and less toxic antibiotic substitutes, physicians are forced to reconsider the use of colistin in many healthcare institutions across the globe, highlighting the pressing need for new antimicrobial medicines [6,7,8]. According to the Infectious Diseases Society of America (IDSA) document “Bad Bugs, No Drugs”, “as antibiotic discovery stagnates, a public health crisis brews” [9].

This article aims to address the escalating threat of colistin resistance, driven by its increasing use as a last-resort antibiotic and the rapid spread of plasmid-mediated resistance genes. With few new antibiotics in development and growing reliance on colistin to combat multidrug-resistant Gram-negative bacteria, the urgency to understand resistance mechanisms and explore innovative solutions is critical. A literature study was conducted to examine global antibiotic resistance, focusing on colistin resistance, AMU, AMR, and stewardship. Databases like Google Scholar, PubMed, and Scopus were searched with relevant keywords. This article provides a comprehensive overview of colistin resistance by detailing its underlying mechanisms, clinical implications, and global epidemiology. It addresses a critical gap in integrating surveillance data with clinical outcomes and highlights the limited awareness of resistance detection in routine diagnostics. By evaluating current strategies to combat resistance and exploring future therapeutic and diagnostic innovations, the study offers a focused roadmap for addressing this urgent public health threat.

## 2. Mechanism of Colistin Resistance

Until the past decade, efforts to combat resistance have primarily focused on research understanding the genetic processes underpinning it (Figure 1). Figure 1 depicts that the colistin resistance in Gram-negative bacteria arises from lipid A modifications via L-Ara4N and pEtN additions. These are regulated by activated PmrA-PmrB and PhoP-PhoQ systems, with pmrCAB and arnBCADTEF operons driving modification synthesis. Mutations in *mgrB* overactivate PhoP-PhoQ, enhancing resistance. The existence of the *MCR-1* gene, which codes for a protein that alters the bacterial cell membrane and lessens the efficiency of colistin, is one important discovery (Figure 2). Figure 2 explains the various mechanisms by which bacteria develop resistance: chromosomal mutation, plasmid borne resistance, and lipid A mutation, which further enhances the unbinding of colistin to the membrane. This gene can spread across bacteria, which exacerbates the issue even more. It has been identified in a number of bacterial species, including *E. coli* (Table 1). Table 1 represents the various mechanism responsible for colistin resistance. Colistin resistance occurs through lipid A modification in the bacterial outer membrane, reducing drug binding. This is regulated by chromosomal PmrAB and PhoPQ two-component systems or plasmid-mediated *MCR* genes encoding phosphoethanolamine (pEtN) transferases.

### 2.1. Polymyxin Resistance Mediated by Plasmids

Plasmid-mediated polymyxin resistance poses a substantial barrier to the treatment of multidrug-resistant GNB, particularly those involving *MCR 1–10* genes. Horizontal gene transfer is facilitated by plasmids, which spreads resistance in environmental and clinical contexts. This resistance poses a special risk in infections that are resistant to carbapenem, as it may result in treatment failures and unfavorable patient outcomes. The creation of substitute therapies, improved monitoring, and strict antimicrobial stewardship are all necessary to address this problem. To stop it from spreading and keep polymyxin medicines effective, it is essential to comprehend the dynamics of this resistance and how it affects treatment [4].

#### 2.1.1. *MCR-1* Gene

The *MCR-1* gene, a plasmid-mediated colistin resistance determinant, has gained significant attention due to its role in conferring resistance to one of the last-resort antibiotics, colistin in GNB. *MCR-1*, which was identified in China in 2015, encodes an enzyme called phosphoethanolamine transferase that alters the lipid a part of the outer membrane of bacteria that inhibits the binding and antibacterial activity of colistin. Because of its placement on plasmids, the gene can move horizontally between bacterial strains and species, which aids in the spread of colistin resistance around the world. The emergence of *MCR-1* produces a serious threat to public health, necessitating enhanced surveillance, antimicrobial stewardship, and the development of alternative treatment strategies to combat the spread of colistin-resistant bacteria [10].

#### 2.1.2. *MCR-2* Gene

In 2016, the *MCR-2* gene was found in Belgium, which revealed a transferase enzyme similar to *MCR-1* that codes for a plasmid-mediated colistin resistance determinant. Through alterations to the lipid A layer of the bacterial outer membrane, the resistance mechanism is similar to *MCR-1* but genetically different; it reduces the effectiveness of colistin, an essential antibiotic of last resort. The rise in *MCR-2* is indicative of the growing danger of colistin resistance, underscoring the need for close observation and rigorous management practices [11,12]. Following its discovery in *E. coli* isolated from cattle and pigs in Belgium [11], the *MCR-2* was most recently found in human vaginal swabs from China [13]. The *MCR-2* (1617 bp) and the amino acid identities of the PEtN transferases encoded by the genes *MCR-1* (541 aa) and *MCR-2* (538 aa), are 81% similar to each other. They are 63% and 64% similar to to *Paenibacillus sophorae* and *Moraxella osloensis*, respectively, according to their phylogenetic research [14].

#### 2.1.3. *MCR-3* Gene

Similar to earlier discovered *MCR*, the *MCR-3* gene is a plasmid-mediated colistin resistance determinant that contributes to colistin resistance by similar transferase enzyme. Initially discovered in 2017 in an *E. coli* isolate from pig in China, *MCR-3* exhibits genetic divergence from *MCR-1* and *MCR-2*, suggesting independent acquisition and dissemination. This gene’s presence on mobile genetic elements facilitates its horizontal transfer between bacterial strains and species, posing a significant challenge to antimicrobial therapy and public health [14].

#### 2.1.4. *MCR-4* Gene

The *MCR-4* gene is a plasmid colistin resistance determinant, similar to other *MCR* variants, leading to resistance against colistin. Initially identified in 2017 in an *E. coli* isolate from Belgium, *MCR-4* exhibits distinct genetic features from previously described *MCR* variants, suggesting independent acquisition and dissemination. Comprehending the genetic traits and prevalence of *MCR-4* is crucial in order to execute efficient monitoring tactics and formulate focused interventions aimed at curbing the proliferation of colistin resistance and maintaining the effectiveness of this vital antibiotic [15].

#### 2.1.5. *MCR-5* Gene

The discovery of the *MCR-5* gene in 2017, found in a d-tartrate fermenting *S. enterica* subsp. isolated from a pig in China, underscores the plasmid-mediated mechanism of colistin resistance. Similar to other *MCR* variations, *MCR-5* also encodes a transferase enzyme, leading to alterations in the lipid A of the bacteria and reducing colistin binding affinity. Its presence on mobile genetic elements facilitates horizontal transfer among bacterial populations, heightening concerns over colistin resistance dissemination. To safeguard the efficacy of this crucial antibiotic in clinical settings, it is imperative to comprehensively characterize the genetic profile and distribution patterns of *MCR-5* to implement effective prevention strategies against colistin-resistant bacteria [16,17].

#### 2.1.6. *MCR-6* Gene

A plasmid-originated pathway of colistin resistance is suggested by the *MCR-6* gene, which codes for an enzyme transferase that is comparable to other *MCR* variants. *MCR-6*, which was identified in 2017 in an *E. coli* isolate from a patient suffering from a bloodstream infection in Switzerland, modifies the lipid A constituent of the bacterial outer membrane to give colistin resistance. Antimicrobial therapy and public health are substantially challenged by its presence on mobile genetic elements, which promotes horizontal transfer between bacterial strains and species. To combat the rise of colistin resistance and maintain the effectiveness of this vital antibiotic, it is imperative to comprehend the genetic makeup and epidemiology of *MCR-6* in order to design focused interventions and surveillance plans that are effective [18].

#### 2.1.7. *MCR-7* Gene

Understanding colistin resistance has advanced significantly with the 2018 discovery of the *MCR-7* gene in *K. pneumoniae* isolates from China. Because it has the ability to transmit genes horizontally through mobile genetic elements, *MCR-7* encodes an enzyme that imparts resistance to colistin, hence posing a severe threat to public health. To effectively apply targeted treatments to minimize its transmission and preserve viable treatment choices for multidrug-resistant infections, it is imperative to have a thorough understanding of the genetic features and epidemiology of *MCR-7* [19].

#### 2.1.8. *MCR-8* Gene

An important addition to the range of plasmid-associated colistin resistance mechanisms is the *MCR-8* gene. *MCR-8* was identified in 2019 in *K. pneumoniae* isolates from China that produce New Delhi metallo-β-lactamase (NDM). Because *MCR-8* can spread horizontally between bacterial strains and species through mobile genetic components, its introduction significantly complicates the AMR landscape. In order to combat the emergence of colistin-resistant bacteria and maintain the efficacy of this crucial antibiotic for clinical use, *MCR-8*’s presence emphasizes the urgent need for improved surveillance, antimicrobial stewardship, and the creation of alternate therapeutic approaches [20].

#### 2.1.9. *MCR-9* Gene

The *MCR-9* gene represents a novel addition to the growing repertoire of colistin resistance mechanisms. Identified in 2019 in clinical isolates of *K. pneumoniae* from China, *MCR-9* also produces an enzyme responsible for conferring colistin resistance. The discovery of *MCR-9* underscores the ongoing evolution of AMR and highlights the urgent need for robust surveillance and control measures to curb its spread. Given its potential for horizontal transfer between bacterial strains and species via mobile genetic elements, the emergence of *MCR-9* further complicates efforts to combat multidrug-resistant infections. To mitigate its spread and maintain the efficacy of colistin for therapeutic use, specific therapies that consider the genetic features and epidemiology of *MCR-9* are essential [21].

#### 2.1.10. *MCR-10* Gene

A recent finding in the field of plasmid-mediated mechanisms of colistin resistance is the *MCR-10* gene. *MCR-10*, which encodes transferase enzyme, was discovered in 2020 in clinical samples of *E. coli* from China. *MCR-10’s* appearance contributes to the increasing complexity of AMR, creating serious problems for clinical care and public health. The spread of *MCR-10* highlights the critical need for improved surveillance, antimicrobial stewardship, and the creation of alternative therapeutic approaches in order to effectively combat multidrug-resistant infections. Understanding the genetic characteristics and epidemiology of *MCR-10* is essential for informing targeted interventions aimed at mitigating its dissemination and preserving the effectiveness of colistin for clinical use [22]. A phylogenetic tree of *MCR-1* to *MCR-10* genes can offer valuable insight into the evolutionary relationships and sequence diversity among phosphoethanolamine transferase enzymes responsible for colistin resistance. Despite sharing a common mechanism of the modification of lipid A, these *MCR* variants exhibit varying levels of sequence identity, influencing their enzymatic efficiency and dissemination potential. A comparative phylogenetic analysis not only highlighted conserved and divergent regions but also aided in identifying specific molecular markers for diagnostic and surveillance purposes. This approach can bridge gaps in understanding the structural and functional diversity underlying plasmid-mediated colistin resistance. We incorporated a phylogenetic tree comparing *MCR-1* to *MCR-10* all sequence collected from NCBI, which highlights the sequence variability among these phosphoethanolamine transferases (Figure 3). Figure 3 illustrates the evolutionary relationships among the *MCR* gene variants based on sequence alignment. Distance values reflect the degree of sequence divergence, with closely related variants such as *MCR-5* to *MCR-8* clustering together, while more divergent forms like *MCR-9* form separate branches. This phylogenetic analysis provides insight into the genetic variability and potential functional differences underlying plasmid-mediated colistin resistance mechanisms.

## 3. Clinical Implication

The emergence and proliferation of colistin resistance present formidable challenges in clinical management and patient care. Gram-negative pathogens like *A. baumannii*, *K. pneumoniae*, and *E. coli,* already notorious for multidrug resistance, now pose an even greater threat due to dwindling treatment options. Once considered a last resort, colistin now struggles against these resistant strains, leading to treatment failures, prolonged hospitalizations, and increased mortality rates. According to Kollef et al., in a study among 93 patients, the mortality rate was 29.9% [23], and 41% had diabetes as a co-morbid condition. *K. pneumoniae* was the most common organism isolated, accounting for 91.4% of cases [23]. In another study, colistin-resistant MDR GNB was found in 19.6% of cases, with *K. pneumoniae* showing 9.2% resistance. Most isolates were from respiratory infections (28.8%) and neurology ICU (54.1%), with a 74.2% recovery rate. Among 122 patients with *A. baumannii* infection, 14.8% had colistin-resistant isolates. The 14-day mortality rate was 44.4% for colistin-resistant cases and 34.6% for colistin-susceptible cases, with no significant difference. *A. baumannii* was a key risk factor for mortality [24]. In Balkhair et al.’s study among 585 bloodstream infections, carbapenem resistance was seen in 27.7% of isolates, with *A. baumannii* (80.4%), *K. pneumoniae* (46.4%), and *P. aeruginosa* (29.9%) being the most resistant. Colistin resistance was found in 13.4% of carbapenem-resistant isolates. The 30-day mortality rate was 68.1% for carbapenem-resistant infections versus 21.3% for carbapenem-susceptible cases [25]. In another study, among 77 patients, those with *A. baumannii* resistant to colistin (ABCR) infections had higher exposure to antibiotics, mechanical ventilation, and invasive procedures compared to *A. baumannii* sensitive to colistin (ABCS) infections. Prolonged carbapenem use was an independent risk factor for ABCR infection [26]. Out of 3190 patients, 54 (1.6%) developed multidrug-resistant urinary tract infection (MDR-UTI), with *P. aeruginosa* being the most frequent pathogen (63%). Colistin was effective in 66.7% of cases, and MDR-UTI patients had a significantly longer hospital stay (9.28 days) [27]. A therapeutic challenge for clinicians caring for critically ill patients hospitalized for acute infections is the rising prevalence (Table 2) of MDR GNB mortality. Table 2 highlights the rising prevalence of colistin-resistant infections caused by various multidrug-resistant microorganisms, posing a significant therapeutical challenge associated with clinicians managing critically ill patients. Mortality rates associated with these infections vary widely, from 2.15% for *K. pneumoniae* in the United Kingdom to 44.4% for *A. baumannii* in Taiwan [23]. This is due to the fact that the potential existence of these organisms raises the possibility that improper initial antimicrobial therapy may lead to treatment failures, poor patient outcomes, and increased medical expenses (Table 3). Table 3 underlines the clinical implication of colistin resistance. Overuse of empirical broad-spectrum antibiotics poses a significant risk of antimicrobial drug resistance, potentially leading to unfavorable patient outcomes [23]. It is crucial to promptly administer optimal antimicrobial therapy to hospitalized patients with acute infections to achieve favorable outcomes and minimize the risk of resistance development, thereby reducing morbidity and mortality. The use of colistin increased significantly in the late 1980s among patients with ventilator-associated pneumonia, convulsions, chemical meningitis, bacteremia, multidrug-resistant GNB, cystic fibrosis, and chemical ventriculitis. This underscores the significance of using antibiotics sparingly when treating a variety of clinical conditions [28,29,30].

## 4. Epidemiology

Colistin resistance has emerged in a number of nations (Table 4) across the world as a result of the increased utilization of colistin for MDR GNB infections. The colistin susceptibility of multidrug-resistant *A. baumannii* from the Western Pacific region was investigated in 2008 as part of the SENTRY antimicrobial surveillance program. The results showed that, with the exception of one isolate, which had a MIC of 128 mg/L, all of the isolates were extremely multi-resistant and had colistin MICs of 0.5–2 mg/L [3,34]. The prevalence of colistin resistance in different countries (Table 5) and in different states in India (Table 6) further indicates raising concern for drug-resistant strains.

### 4.1. American Countries

The resistance rates for *A. baumannii* and *P. aeruginosa* comprise less than 5.5% of the data from the US; however, the resistance rates for *K. pneumoniae* are greater. Despite combining oral ciprofloxacin and nebulized colistin–methate sodium, Denmark has not seen the development of colistin resistance in multidrug-resistant *P. aeruginosa* [59]. In 2007–2008, colistin showed efficacy (MIC90, ~2 mg/L) against a range of Gram-negative bacilli in Canadian hospitals, with MDR *P. aeruginosa* remaining sensitive [59]. Four percent of the isolates in an Argentine ICU, mostly from clones I and III, had colistin hetero-resistance. *P. aeruginosa* and *A. baumannii* resistance rates reached 9% in South America. In Band et al.’s 2021 study, 69.2% colistin resistance was discovered in the United States alone [35,60].

### 4.2. European Countries

The prevalence of colistin resistance varies greatly throughout Europe, as several studies have shown worrying patterns. 97% of *P. aeruginosa* isolates in the UK were found to be colistin-sensitive [60], while 3.1% were resistant, according to a survey of CF patients. Overall, 19.1% of *A. baumannii* isolates in Spain tested positive for colistin using the broth microdilution technique and the E-test. Resistance rates of 17% for *Klebsiella* spp. and 11% for *E. coli* were recorded in Romania [61]. Overall, 36.1% of carbapenem-resistant *K. pneumoniae* isolates were found to be colistin-resistant in a Rome, Italy study, which was associated with higher medication use. Colistin-resistant *K. pneumoniae* increased from 1% in Greece in 2005 to 19% in 2008 [62]. Data from Hungary in 2008–2009 showed that isolates of *K. pneumoniae* with several resistance genes had a high level of colistin resistance. Furthermore, epidemiological research conducted in Europe reported a substantial use of colistin in animal health [63].

### 4.3. African Countries

Resistant rates to colistin were found to be less than 10% in studies from South Africa and less than 10% in those from Nigeria. Overall, 121 strains of *Enterobacteriaceae* resistant to colistin were recovered from 93 individuals by Mezghani Maalej et al. of Tunisia in retrospective research. According to the study, the percentage of *E. coli, K. pneumoniae*, and *E. cloacae* that were resistant to colistin varied from 0.09% to 1.2% and 1.5%, respectively [64]. Yet, cumulative data for *P. aeruginosa* from two distinct institutions in Zimbabwe revealed 53% resistance rates [65]. Recent findings have highlighted the geographical distribution of specific sequence types (STs), such as ST131, which is widely prevalent in Europe and Asia. ST410 has also emerged globally with increased carbapenem resistance. In Africa, ST167 and ST405 are frequently detected in clinical isolates. These patterns underscore the regional variation in resistance-associated STs [66].

### 4.4. Asian Countries

Studies carried out in the Asia-Pacific region demonstrate that different *Enterobacter* species and *Klebsiella* species have different prevalence of colistin resistance, with hetero-resistance patterns being prevalent. Overall, 0.3% of *Klebsiella* species and 21% of *Enterobacter* species were found to be resistant to colistin; resistance rates varied from 13.8% in India to 50% in the Philippines [34,67,68]. All strains of *Acinetobacter* and *E. coli* in Singapore remained susceptible to colistin, whereas 30% of *P. aeruginosa* isolates showed resistance to the antibiotic. According to a different Chinese investigation, all *A. baumannii, A. calcoaceticus* complex isolates were vulnerable to polymyxin, although only 8% of them were resistant to carbapenems. In another study done in China by Huang et al., 2020, colistin resistance was 32.7%. In Japan, 7.7% colistin resistance cases was discovered in 2022; in India, the colistin resistance percentage was found to be 1.28%; and in Thailand, the colistin resistance percentage was found to be 3.3% alone, further rising the concern for antimicrobial resistance [36,37,38,39,67].

## 5. Surveillance

The recommendation of a phenotypic screening method (Figure 4) for colistin resistance identification in a resource-constrained situation is challenging. Figure 4 explains about surveillance methods for colistin resistance with various methods in clinical and environmental settings like phenotypic testing, genotypic testing, research and development related to new therapeutics, multinational collaboration, and the development of rapid diagnostic tools. In order to handle the technicalities of colistin susceptibility testing with standard controls and procedures to monitor the environmental samples and food-producing animals, there should be at least one standard surveillance laboratory at the national level (Table 7). Table 7 highlights the various surveillance methods for colistin resistance. Effective identification of *MCR*-mediated colistin resistance necessitates a multimodal strategy that combines selective agar plates such as COL-APSE or CHROMID^®^ Colistin R agar with PCR-based methods for *MCR* gene identification. Innovations in rapid culture-based testing are essential to accelerating identification procedures. To increase colistin resistance testing throughput, accuracy, and speed, improved automated Antimicrobial Susceptibility Testing (AST) systems are necessary. As a timely and accurate detection approach, the Rapid Polymyxin NP test shows promise in reliably differentiating between susceptible and resistant strains [69,70]. Innovations in rapid, culture-based testing are essential for accelerating identification procedures, especially in settings with limited resources. Improved automated AST systems are also necessary to increase throughput, accuracy, and speed in colistin resistance testing like VITEK 2. The Rapid Polymyxin NP test has emerged as a promising tool for the timely and accurate detection of colistin resistance, reliably differentiating between susceptible and resistant strains. This method could be particularly beneficial in resource-constrained environments where rapid decision-making is crucial [71].

Global health organizations play a vital role in standardizing surveillance and reporting methods for colistin resistance. By providing guidelines, training, and support, these organizations can help ensure that monitoring programs are consistent, accurate, and effective across different regions. Standardized reporting methods would facilitate the comparison of data on a global scale, enabling the better tracking of colistin resistance patterns and more collaborative responses. In resource-limited settings, international collaborations and support from global health organizations are essential for building capacity, providing necessary resources, and ensuring that even the most constrained environments can contribute to the global effort to monitor and combat colistin resistance. Consideration of recent advancements in genome-based rapid detection tools could further revolutionize colistin resistance diagnostics. These methods leverage next-generation sequencing and other molecular approaches to rapidly and accurately detect resistance genes [72]. However, their implementation in resource-limited settings poses challenges, such as high costs, the need for specialized training, and infrastructure requirements. Addressing these barriers is vital to ensuring global applicability and impact. International collaborations and capacity-building initiatives, supported by global health organizations, could facilitate the adoption of these technologies even in constrained environments. By overcoming these hurdles, genome-based tools could significantly enhance the precision and efficiency of colistin resistance surveillance, contributing to a more coordinated global response.

### 5.1. The Initial Screening Process Using Selective Culture Medium

Novel selective culture media, like chromogenic variants, are pivotal in clinical microbiology, especially for detecting infections in high-risk groups such as recent hospital admissions or those with antibiotic exposure. These media aid in implementing tailored infection control measures and hold promise for community-acquired infections pending further validation. Additionally, their application extends to veterinary surveillance, where screening for antibiotic resistance in animal bacteria is becoming imperative. The advent of Super Polymyxin selective medium showcases significant advancements in identifying polymyxin-resistant Gram-negative isolates with high sensitivity and specificity, facilitating prompt and accurate diagnosis for optimized treatment strategies [73,74,75].

### 5.2. Automated AST, Antibiotic Gradient Testing, and Disk Diffusion Assays

The limitations of disk diffusion and gradient testing for colistin resistance highlight the need for rapid, reproducible culture-based AST. Broth microdilution (BMD) is currently the most accurate method, but it is time-consuming [76]. Accurate colistin Minimum Inhibitory Concentration (MIC) testing is crucial due to its significant side effects. A cost-effective and globally applicable solution is the colorimetric test, providing quick results within four hours, crucial for timely treatment decisions, especially in cases of bacteremia and sepsis amidst multidrug resistance challenges [76]. The importance of rapid phenotypic test confirmation of resistance has been highlighted by Jayol et al., especially if it can be carried out straight from blood culture broth instantaneously and enable early antimicrobial medication adaptation in the event of multidrug resistance [77].

### 5.3. Molecular Diagnostics

While MALDI-TOF MS shows potential for identifying specific proteins, directly detecting the *MCR* protein still remains elusive. Advancements in Mass Spectroscopy technology may enable future tests to find out enzymatic activity or altered LPS molecules [74]. However, pinpointing resistance via LPS-modifying enzymes poses challenges, as resistance can stem from subtle alterations. Molecular identification of *MCR*-like genes via PCR is straightforward, aided by rapid real-time techniques [78]. Yet, complete diagnosis may be hindered by co-occurring resistance mechanisms, including amino acid changes in regulatory proteins. Connecting mutations to susceptibility alterations requires extensive sequencing, highlighting the complexity of resistance detection [79].

## 6. Combating Colistin Resistance

In acute care and hospital settings, strict infection control protocols are vital to curb the spread of MDR GNB. This involves promptly identifying resistant infections, educating healthcare staff on hand hygiene, and adhering to Clinical and Laboratory Standards Institute (CLSI) standards for early resistance detection. Surveillance cultures should be conducted regularly until no new cases emerge, with prompt notification to infection control specialists upon detection of resistant organisms. Robust antimicrobial stewardship programs are crucial for effective therapy, optimizing antibiotic regimens and reducing hospital stays through quicker positive blood culture notifications and timely medication administration [80].

### 6.1. Colistin Combination Therapy

It is necessary to combine colistin with other antibiotics due to the growing resistance to it. Colistin and other medications can work in synergy, according to studies. Sorli et al.’s study, for instance, demonstrated that imipenem and colistin worked synergistically to combat *K. pneumoniae*, but only 11% of isolates that were resistant to colistin exhibited this synergy. Colistin-heteroresistant bacteria frequently arise with colistin-only treatment. Overall, 22 patients with MDR *A. baumannii* pneumonia received intravenous treatment with colistin and rifampicin in another trial by Bassetti et al.; the patients had good results with few side effects [81,82,83].

Similarly, Song et al. studied 10 patients suffering from ventilator-associated *pneumonia* treated with colistin and rifampicin, demonstrating the safety and efficacy of combination therapy. While reports on the efficacy of combination therapy versus monotherapy have been conflicting, some studies have shown a higher clinical cure rate and microbiological eradication rate with combination therapy, though not statistically significant [80,84]. Data indicate that combinations of ciprofloxacin, piperacillin, aztreonam, imipenem, rifampicin, and arbekacin are both safe and effective, with colistin and rifampicin demonstrating the highest synergy. Additionally, combinations containing tigecycline, carbapenems, and gentamicin have shown varying degrees of synergy. While colistin and vancomycin together significantly increase the risk of renal failure in critically sick patients with carbapenem-resistant *A. baumannii* infections, the clinical outcomes are comparable whether vancomycin is used or not [85,86].

### 6.2. One Health Perspective

The overuse of antibiotics in the human, animal, and environmental sectors has led to the urgent worldwide concern known as AMR. The widespread usage of colistin in animal husbandry, especially in China and Southeast Asia, has accelerated the emergence of *Enterobacteriaceae* resistant to colistin, making the treatment of multidrug-resistant infections more difficult. A multimodal strategy including enhanced antimicrobial policies, surveillance, stewardship, infection control, and sanitation measures is needed to address AMR [86,87]. The need for concerted measures to protect medically important antibiotics is highlighted by international programs like the WHO Global Action Plan on AMR and the US National Action Plan for Combating Antibiotic-Resistant Bacteria. In order to stop the spread of resistant diseases, countries need to strengthen their monitoring capacities. This can be done with the help of the WHO’s Global AMR monitoring System (GLASS) [88,89]. New research from a country in South America has emphasized the value of infection control and stewardship measures in stopping the spread of high-risk clones that cause AMR. Developing comprehensive policies to reduce the deleterious effects of AMR on global health, security, and the economy requires stakeholder engagement [90,91].

## 7. Future Perspective

Carbapenem-resistant GNB develop colistin resistance through mechanisms like plasmid-mediated LPS modifications and mutations such as LAra4N. Despite its bactericidal action, the exact mechanism of colistin’s activity remains unclear, emphasizing the need for further research. Spontaneous resistance raises concerns about its efficacy against multidrug-resistant infections, making it critical to understand and address these resistance mechanisms to prevent widespread dissemination [90]. Fluopsin C, a metal-containing antibiotic from *Streptomyces* and *Pseudomonas* species, exhibits potent activity against Gram-negative, and drug-resistant bacteria. Terrein, derived from fungi, is effective against *S. aureus*, *A. hydrophila*, and *E. faecalis*, while ellipticine, known for anticancer properties, shows antibacterial activity against colistin-resistance strain of *E. coli*. Emerging strategies like repurposing drugs (e.g., niclosamide, PFK-158), CRISPR-based approaches, nanotechnology, and photodynamic therapy (PDT) enhance colistin’s efficacy. Phage therapy combined with colistin offers a promising avenue to mitigate resistance and treat multidrug-resistant infections [92].

Future perspectives in colistin resistance research are poised to address the pressing need for effective strategies against multidrug-resistant bacteria (Figure 5). Figure 5 depicts the potential strategies and innovations for future perspectives in colistin resistance research for novel therapeutic approaches like vaccination therapy, antimicrobial peptide, phage therapy, antibodies used as drugs, and synergy treatment. Investigating the molecular mechanisms of colistin resistance is essential to counteracting it. Effective management requires a One Health approach that takes into account environmental, animal, and human issues. Thorough surveillance in environmental and clinical contexts can offer important insights into resistance epidemiology and help direct management strategies. Emerging technologies like genomics and bioinformatics are playing a pivotal role in understanding and combating colistin resistance. Genomic sequencing allows for the detailed analysis of resistance mechanisms at a molecular level, enabling the identification of specific genes, such as the *MCR* gene, responsible for colistin resistance [93]. Bioinformatics tools can analyze vast datasets to track the evolution and spread of resistance genes across different regions and environments. These technologies are crucial for developing targeted interventions and guiding the use of colistin in clinical settings. However, the effective deployment of these tools requires robust policy frameworks and international cooperation.

Regarding the of policy and international cooperation, policymakers must prioritize investment in these technologies and collaborate globally to share data and standardize approaches to surveillance. International organizations like the WHO play a critical role in coordinating efforts to address antimicrobial resistance, ensuring that resources and knowledge are shared across borders to combat this global health threat [94].

### 7.1. Contemporary Instruments for Quick AMR Diagnosis

Outpatient clinics urgently require rapid and decentralized diagnostics to mitigate antibiotic overuse, a pressing issue at present. It is imperative to swiftly identify the causative pathogen, differentiate between bacterial and viral infections, detect microbial antibiotic resistance, and ascertain the most suitable antimicrobial treatment. By doing so, the over prescription of antibiotics can be minimized, and the emergence of antibiotic resistance can be more effectively managed [95].

Global health threats in the 21st century are deemed to be greatest by the WHO, who calls for immediate response. For resistance testing, prompt diagnosis is essential, particularly in areas with high AMR rates. The current AST standards require the assessment of antimicrobial effectiveness using pure culture isolates; nevertheless, it might be difficult to differentiate between benign and dangerous species. The misuse of empirical treatment continues despite advancements in diagnosis [96].

### 7.2. Available Technologies for Rapid AST

Though labor-intensive stages like enrichment cultures and pure culture separation are frequently needed, several approaches promise fast AST within minutes or hours. In theory, non-purified polymicrobial samples might be used with Nucleic Acid Amplification Tests (NAAT)-based techniques, nucleic acid hybridization, or immunodiagnostics. NAAT can be used to determine MIC and detect AR after a brief antibiotic culture. Some very sensitive immunodiagnostics track growth in real time, but most quick AST techniques rely on endpoint analysis. Clinical studies using biosensor technology for motility, thermal changes, or microbial metabolism are, nevertheless, scarce. Outpatient clinics have limited access to inexpensive, dependable, quick, and easy-to-use AST devices [97].

### 7.3. NAAT

Aided by diagnostic panels from firms such as BioMérieux, Elitech, and Qiagen, which may detect specific AR genes, while their presence alone does not always suggest resistance, NAAT are particularly successful in syndromic detection of infections. NAAT are unable to provide treatment plans or establish MICs, but they can update for new resistance indicators. MIC values can be estimated by qPCR and qRT-PCR, but the latter is still expensive. Enrichment cultures and nucleic acid purification are frequently required for precise identification, reducing impurities and template loss. While advances like T2Biosystems’ magnetic resonance panel and molecular beacons in real-time PCR highlight important advancements in resistance gene detection from blood, DNA array hybridization improves sensitivity and specificity [98].

### 7.4. Whole Genome Sequencing

WGS has progressed to a stage where it serves as a viable method for identifying pathogens and determining AST. Third-generation sequencing platforms capable of swiftly providing reasonably long reads include the Oxford Nanopore MiniON and PromethION, the Illumina MiniSeq, and the PacBio Sequel system from Pacific Biosciences. WGS holds the promise of simultaneously offering rapid pathogen identification, epidemiological typing, and the detection of medication susceptibility genes. However, interpreting WGS results, which yield a vast amount of fragmented data, necessitates sophisticated algorithms [99]. Incorporating whole-genome sequencing (WGS) into research provides a powerful tool for uncovering cryptic resistance genes that may evade detection by conventional diagnostic methods. Although not yet routinely implemented in clinical settings, WGS enables the high-resolution identification of both known and novel resistance determinants, including those expressed at low levels or embedded within complex genomic contexts. This approach not only enhances our understanding of the genetic basis of colistin resistance but also supports the development of curated resistance gene databases. Furthermore, WGS can reveal insights into the pathogenicity and virulence potential of bacterial isolates, offering a broader perspective on their clinical impact. As such, the use of WGS represents a foundational strategy for academic institutions and surveillance programs seeking to stay ahead of emerging resistance threats. For instance, a study on *Citrobacter werkmanii* utilized WGS to uncover mutations in the pmrA gene, leading to acquired colistin resistance. The research highlights the potential of WGS in identifying resistance mechanisms that are not immediately apparent through conventional testing methods. Such findings underscore the importance of integrating WGS into surveillance efforts to monitor and combat antimicrobial resistance effectively [100].

The European Committee for Antimicrobial Susceptibility Testing assessed WGS’s potential for AST in 2017. Since there was not enough information to use WGS as an AST tool for most bacteria, they came to the conclusion that it was not appropriate for clinical decision-making. To make system and bioinformatics tool comparisons easier, they emphasized the vital requirement for a single database that contains all known resistance genes and mutations [101].

## 8. Discussion

Colistin, a treatment for MDR infections of Gram-negative bacteria, is increasingly facing resistance due to both chromosomal mutations and plasmid-borne MCR genes [102]. The growing prevalence of colistin resistance is particularly concerning given its vital role in treating severe infections where other antibiotics fail. The mechanisms driving this resistance, the clinical implications, and the global efforts to combat its spread represent significant challenges for healthcare systems worldwide. This review delves into the current understanding of colistin resistance and the multidimensional strategies necessary to mitigate its impact on global health [103].

Colistin resistance arises from two primary mechanisms: chromosomal mutations and plasmid-mediated resistance genes. Chromosomal mutations, especially those affecting the bacterial outer membrane’s lipid A component, alter the binding affinity of colistin, rendering it less effective; for example, a study highlighted the emerging colistin resistance in *A. baumannii* isolates from ICU patients in Bhubaneswar, India, with 7% showing resistance. The findings emphasize the urgent need for continuous surveillance and alternative treatment strategies to combat multidrug-resistant infections [104]. These mutations modify the lipid A structure by adding molecules such as phosphoethanolamine (pEtN), which reduces the drug’s ability to interact with the bacterial cell membrane d-mediated resistance, driven by *MCR* genes (*MCR-1* through *MCR-10*), posing a particularly significant threat due to its potential for horizontal gene transfer between bacterial species [105]. *MCR-1*, the most extensively studied gene, was first discovered in *E. coli* isolates from food animals in China in 2015; it has been detected in human clinical isolates, environmental samples, and various animal species across the globe. *MCR-1* encodes a pEtN transferase enzyme that modifies lipid A, similarly to chromosomal mutations, and reduces colistin’s bactericidal activity [106]. The discovery of additional *MCR* variants (*MCR-2* through *MCR-10*) indicates that colistin resistance is evolving rapidly, compounding the complexity of controlling its spread [107].

Colistin-resistant infections are associated with increased morbidity and mortality, particularly in critically ill patients. Infections caused by strains such as carbapenem-resistant *Enterobacteriaceae* (CRE) and *A. baumannii* lead to higher rates of treatment failure, prolonged hospital stays, and the necessity for more toxic or less effective antibiotics. Studies have indicated that mortality rates associated with colistin-resistant infections can exceed 40%, especially when alternative therapeutic options are limited [93].

The rise of colistin resistance puts immense pressure on healthcare systems, particularly in resource-limited settings where diagnostic capabilities and access to alternative treatments are constrained [108]. Clinicians often resort to combination therapies, pairing colistin with other antibiotics like carbapenems or tigecycline, in an attempt to overcome resistance. However, the effectiveness of such regimens in consideration has shown no significant improvement in patient outcomes. Moreover, the nephrotoxicity associated with colistin use can cause further complications in critically ill patients, necessitating careful monitoring and dose adjustments [109].

Colistin resistance is a global phenomenon, with prevalence across different regions. In Europe, resistance rates among *K. pneumoniae* isolates range from 2.1% in the United Kingdom to as high as 59.6% in Greece [31,110]. In the United States, recent studies have reported resistance rates as high as 13.4% among carbapenem-resistant strains. In Asia, colistin resistance is particularly concerning in countries like China, where the increased use of colistin in agriculture has facilitated the emergence and dissemination of MCR genes [111].

The spread of colistin resistance in low- and middle-income countries is further exacerbated by the misuse of antibiotics in the clinical settings. In countries where colistin is still used as a growth promoter in animal husbandry, the risk of transmission from animals to humans is significant; consequently, the integration of human, animal, and environmental health is critical in addressing this issue. Effective surveillance and strict regularization in agriculture are essential for curbing the spread of resistance [112].

The detection of colistin resistance in clinical settings remains a challenge. Current diagnostic methods, such as broth microdilution, are resource-intensive, making them less suitable for routine use in many laboratories. Newer methods, such as the Rapid Polymyxin NP test, offer promise for faster detection of colistin resistance, allowing for more timely interventions [113]. However, these tests are not yet widely available and are only available in co-limited settings where colistin resistance is most prevalent.

Global surveillance efforts, such as those led by the World Health Organization (WHO) in tracking the spread of colistin resistance, are needed. The WHO GLASS has provided a framework for standardized data collection, helping to identify hotspots of resistance and inform public health interventions [114]. However, more comprehensive surveillance systems are needed, particularly in low-income countries where the burden of antimicrobial resistance is highest.

Antimicrobial stewardship programs play a crucial role in reducing the inappropriate use of antibiotics, emphasizing the need for judicious antibiotic use, and ensuring that colistin is reserved for cases where no other therapeutic options are available. Infection control measures, such as stringent hand hygiene practices and environmental cleaning protocols, are also essential in preventing the spread of resistant organisms in healthcare settings [115].

In addition to stewardship and infection control, advance treatments are urgently needed for MDR bacterial infectious diseases. Novel antibiotics targeting colistin-resistant bacteria are in development, but their clinical availability is still several years away. In the meantime, combination involving colistin and other antibiotics, such as rifampicin, have shown promise in some studies, although their efficacy is not guaranteed [116]. Nanotechnology and PDT also offer potential as adjunctive treatments, enhancing activity and reducing the likelihood of resistance [117].

Finally, the development of rapid diagnostic tools and molecular assays for detecting MCR genes is critical for determining colistin-resistant infections early. Whole-genome sequencing (WGS) has emerged as a powerful tool for tracking the evolution and spread of resistance genes, offering insights the mechanisms underlying colistin resistance and guiding public health interventions [117].

## 9. Conclusions

The rapid rise of colistin resistance in Gram-negative bacteria represents a significant public health challenge, demanding immediate and integrated effort. As recent research has revealed, the mechanisms behind colistin resistance are complex, involving both chromosomal mutations and plasmid-mediated genes, complicating efforts to combat this threat. Clinically, the emergence of colistin resistance drastically limits treatment options for infections caused by multidrug-resistant organisms, underscoring the need for alternative therapeutic strategies. Future studies should prioritize understanding resistance dynamics, developing rapid diagnostics, exploring combination therapies, and assessing environmental factors influencing resistance. Vigilant global surveillance is crucial for early detection and containment, while a comprehensive approach that includes antimicrobial stewardship, stringent infection control, and the development of novel treatments such as combination therapy is essential. The gravity of this issue necessitates a concerted effort from researchers, clinicians, and policymakers to invest in further research, develop rapid diagnostic tools, and implement effective strategies to curb the spread of colistin resistance and safeguard the efficacy of antibiotics for future generations.

## Figures and Tables

**Figure 1 antibiotics-14-00958-f001:**
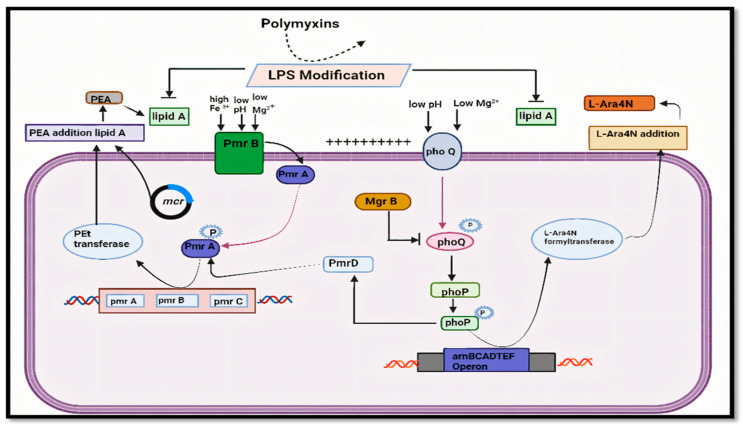
Schematic representation of colistin resistance via lipid A modification regulated by PmrAB, PhoPQ systems, and *MCR* gene-mediated pEtN transfer.

**Figure 2 antibiotics-14-00958-f002:**
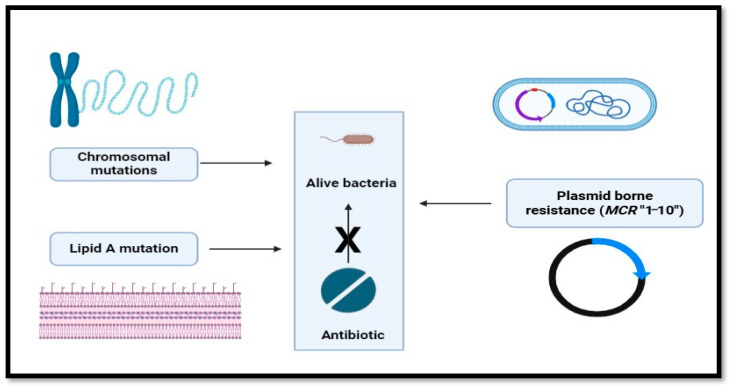
Overview of bacterial colistin resistance mechanisms including chromosomal mutations, plasmid-borne *MCR* genes, and lipid A modifications reducing colistin binding.

**Figure 3 antibiotics-14-00958-f003:**
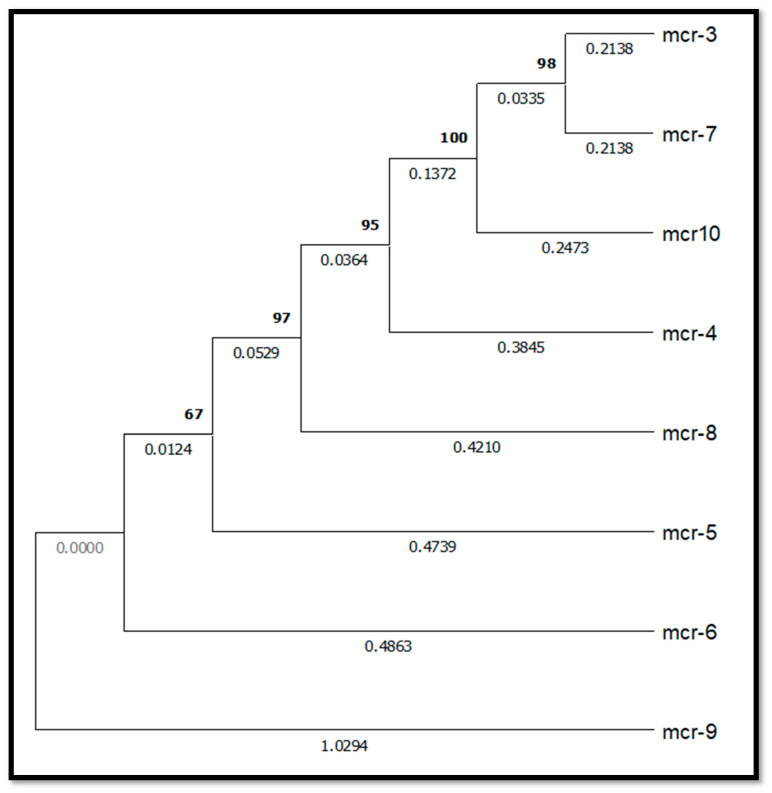
Phylogenetic tree showing the evolutionary relationships among *MCR-1* to *MCR-10* genes based on sequence alignment. Variants cluster according to sequence similarity, highlighting genetic diversity among colistin resistance determinants.

**Figure 4 antibiotics-14-00958-f004:**
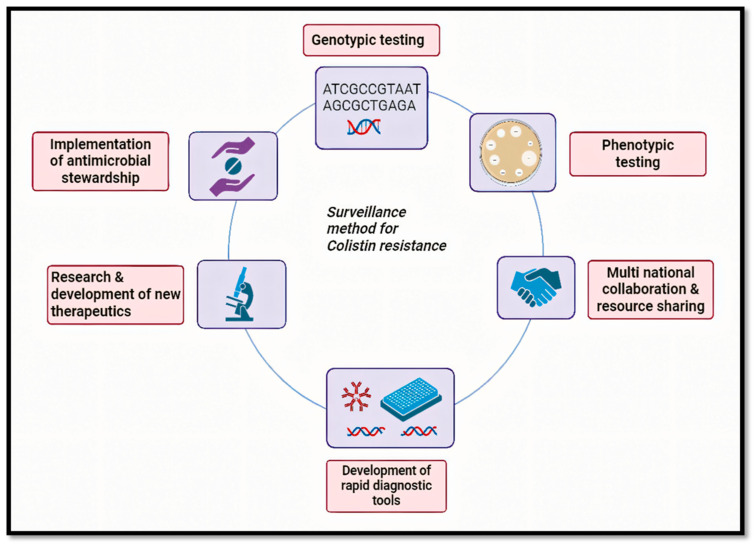
Overview of surveillance methods for detecting colistin resistance across clinical and environmental settings, including phenotypic and molecular techniques.

**Figure 5 antibiotics-14-00958-f005:**
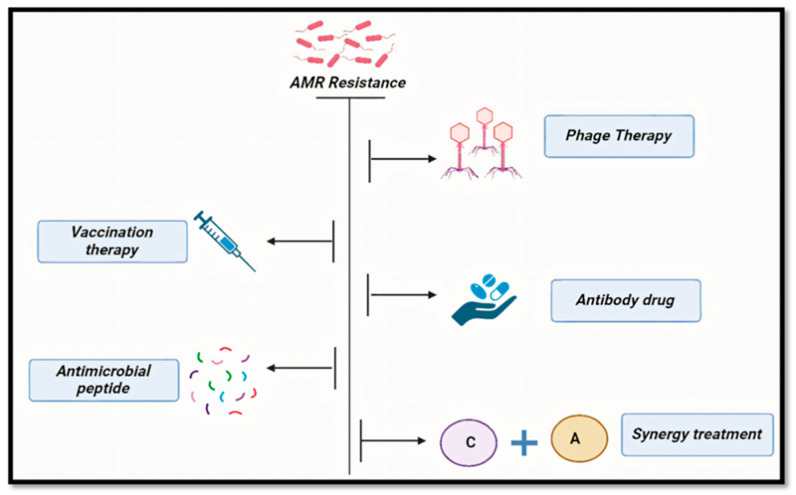
Emerging strategies and innovative approaches for future research in colistin resistance, focusing on novel therapeutics, diagnostics, and resistance mitigation.

**Table 1 antibiotics-14-00958-t001:** Major mechanisms of colistin resistance, including LPS modification, gene mutations, plasmid acquisition, and efflux pump overexpression.

Mechanism	Description
Modification of LPS	Addition of positively charged molecules to lipopolysaccharide (LPS) structure, reducing colistin binding
Mutations in pmrAB	Alterations in regulatory genes controlling LPS modification
Mutations in mgrB	Loss of function mutations in the negative regulator of PhoPQ signaling pathway
Plasmid-mediated mechanisms	Acquisition of mobile genetic elements carrying resistance genes
Efflux pump overexpression	Increased expression of efflux pumps, reducing intracellular colistin concentration

**Table 2 antibiotics-14-00958-t002:** Clinical implications of colistin-resistant gram-negative bacteria in relation to mortality rates.

S.No.	Mortality Rate %	Colistin Resistance Associated Infection (Micro-Organism Name)	Region	Reference
1.	2.15	*Klebsiella pneumoniae*	United Kingdom	[31]
2.	23.1	*Pseudomonas aeruginosa*	Spain	[32]
3.	28	*E. coli*	Turkey	[33]
4.	44.4	*Acinetobacter baumannii*	Taiwan	[24]

**Table 3 antibiotics-14-00958-t003:** Summary of clinical implications associated with colistin resistance in bacterial infections.

Implication	Description
Limited Treatment Options	Reduced efficacy of colistin and polymyxins against resistant strains
Increased Morbidity/Mortality	Higher rates of treatment failure and patient mortality
Spread of Resistance	Potential for dissemination of resistant strains within healthcare settings
Need for Surveillance	Importance of monitoring colistin resistance rates for infection control
Development of Novel Therapies	Urgency for research into alternative treatment options

**Table 4 antibiotics-14-00958-t004:** Prevalence of colistin resistance with associated contributing factors influencing its emergence and spread.

Region	Prevalence of Resistance (%)	Contributing Factors
North America	Low	Stringent antibiotic stewardship practices
Europe	Moderate	Increased use of colistin in agriculture
Asia	High	Widespread use of colistin in human medicine and agriculture
Africa	Varies	Limited surveillance and healthcare resources
South America	Varies	Variable regulatory oversight of antibiotic use

**Table 5 antibiotics-14-00958-t005:** Country-wise prevalence of colistin resistance, highlighting geographic disparities and surveillance data.

Country	Prevalence of Colistin Resistance (%)	References
United States	10.1%	[35]
China	32.7%	[36]
India	1.28%	[37]
Japan	7.7%	[38]
Thailand	3.3%	[39]
South Korea	4.4%	[40]
Russia	30%	[41]

**Table 6 antibiotics-14-00958-t006:** State-wise distribution of colistin resistance in India, reflecting regional trends and reported prevalence rates.

State	Prevalence of Colistin Resistance (%)	Reference
Delhi	15	[42]
Tamil Nadu	33	[43]
Karnataka	1.6	[44]
Jaipur	6.2	[45]
West Bengal	22.5	[46]
Madhya Pradesh	17	[47]
Rajasthan	17.4	[48]
Punjab	12.5	[49]
Haryana	3.8	[50]
Kerala	4.6	[51]
Odisha	13.5	[52]
Jharkhand	9.09	[53]
Chhattisgarh	0	[54]
Uttarakhand	8	[55]
Himachal Pradesh	11.76	[56]
Manipur	2	[57]
Meghalaya	2.5	[58]

**Table 7 antibiotics-14-00958-t007:** Summary of surveillance methods used to detect colistin resistance in clinical and environmental samples, including phenotypic and molecular approaches.

Method	Description
MIC Determination	Minimum Inhibitory Concentration testing to assess bacterial susceptibility to colistin
Whole-genome sequencing	Comprehensive analysis of bacterial genomes for resistance determinants
PCR-based assays	Polymerase chain reaction assays targeting specific resistance genes
Phenotypic Screening	High-throughput screening methods to detect resistant isolates
Epidemiological Surveillance	Monitoring of resistance trends in healthcare settings and communities

## Data Availability

Not applicable.

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
