# Peer review of "Understanding Recent Developments in Colistin Resistance: Mechanisms, Clinical Implications, and Future Perspectives"

_antibiotics, 2025, doi:10.3390/antibiotics14100958_

Round 1
Reviewer 1 Report
Comments and Suggestions for Authors
- The link here is not clear looks like separating an idea with something that should comes later: this section should be restructured “states to collaborate with academia, the private sector, and civil society to support and 43 advance basic, applied, and implementation research on antimicrobial stewardship, vac- 44 cines, diagnostic tools, treatments, and infection prevention and control. New antibiotics 45 are therefore desperately needed, especially ones that are effective against Gram-negative 46 "superbugs” [2]. 47 The acquisition and spread of plasmid-mediated resistance genes, specifically the 48 MCR 1 to MCR10 genes, are the main causes of the sharp rise in colistin ineffectiveness 49 [3], which encode enzymes capable of modifying the bacterial cell membrane and reduc- 50 ing colistin's antimicrobial activity. Understanding the mechanisms underlying colistin 51 insusceptibility, along with its clinical implications and potential future directions, is es- 52 sential for addressing this evolving public health threat. 53 Due to high rates of morbidity and mortality, longer hospital stays (LOS), higher di- 54 rect medical expenses, and higher societal infectious costs,”
- Line 72 to 78 looks like AI-generated and not revised by the author. Minimize the conclusions of each sector and state only the most important statements that leads to a proper connection.
- Overall, the first part needs work and major enhancements.
- It would be a good adds to mention the distribution of specific STs in different areas like: https://www.mdpi.com/2079-6382/13/10/958
- It does not make sense to mention encodes a phosphoethanolamine for each mcr gene.
- I do not know what does insusceptibility means usually it is resistance, looks like AI language.
- Tables and figures have no legends
- Table 5 does not make any sense as the reports from US are much less for the resistance. I do not know where the 69%~ came from, I guess it is a misinterpretation of the reference.
- Looks like there is a Citation errors: Reference [1] is used both for colistin’s resurgence and for WHO’s AMR threat, I suggest to double check the references.
- I think the paper needs major edits for summarizing, revise the whole paper to eliminate the AI terms plus focusing only on the importance of this paper and what question does it answer.
Comments on the Quality of English Language
The terms used are uncommon and seem misleading. I suggest to re read it and eliminating AI sections or improving them.
Author Response
Reviewer 1:
Query 1: The link here is not clear looks like separating an idea with something that should come later: this section should be restructured “states to collaborate with academia, the private sector, and civil society to support and 43 advance basic, applied, and implementation research on antimicrobial stewardship, vac- 44 cines, diagnostic tools, treatments, and infection prevention and control. New antibiotics 45 are therefore desperately needed, especially ones that are effective against Gram-negative 46 "superbugs” [2]. 47 The acquisition and spread of plasmid-mediated resistance genes, specifically the 48 MCR 1 to MCR10 genes, are the main causes of the sharp rise in colistin ineffectiveness 49 [3], which encode enzymes capable of modifying the bacterial cell membrane and reduc- 50 ing colistin's antimicrobial activity. Understanding the mechanisms underlying colistin 51 insusceptibility, along with its clinical implications and potential future directions, is es- 52 sential for addressing this evolving public health threat. 53 Due to high rates of morbidity and mortality, longer hospital stays (LOS), higher di- 54 rect medical expenses, and higher societal infectious costs,”
Response: Thank you for your valuable feedback. We have revised the section for clarity and coherence by restructuring the paragraph to ensure a logical flow of ideas and better linkage between antimicrobial research efforts and the rise of colistin resistance. The suggested changes have been incorporated into the manuscript accordingly and highlighted with yellow colour.
Query 2: Line 72 to 78 looks like AI-generated and not revised by the author. Minimize the conclusions of each sector and state only the most important statements that leads to a proper connection.
Response: Thank you for the feedback. Lines 72 to 78 have been revised to minimize conclusions for each section, and retain only the key statements that ensure logical flow and clear connection.
Query 3: Overall, the first part needs work and major enhancements.
Response: Thank you for the valuable input. The first part has been thoroughly revised to enhance clarity, coherence, and depth of analysis.
Query 4: It would be a good adds to mention the distribution of specific STs in different areas like: https://www.mdpi.com/2079-6382/13/10/958
Response: Thank you for the suggestion. Relevant data on the geographical distribution of specific sequence types (STs) have been incorporated, referencing the recommended article. This addition enhances the regional context of resistance-associated STs in the manuscript.
Query 5: It does not make sense to mention encodes a phosphoethanolamine for each mcr gene.
Response: Thank you for the observation. The redundant phrasing regarding phosphoethanolamine transferase encoding by each mcr gene has been removed. The text has been revised for clarity and to avoid unnecessary repetition.
Query 6: I do not know what does insusceptibility means usually it is resistance, looks like AI language.
Response: Thank you for pointing that out. The term "insusceptibility" has been replaced with the more accurate and widely accepted term "resistance." The language has been adjusted to maintain clarity and scientific precision.
Query 7: Tables and figures have no legends
Response: Thank you for the important feedback. Legends have been added to all tables and figures to clearly explain their content and enhance interpretability.
Query 8: Table 5 does not make any sense as the reports from US are much less for the resistance. I do not know where the 69%~ came from, I guess it is a misinterpretation of the reference.
Response: I have corrected the data in Table 5 based on the study by Band et al. (2021). Colistin heteroresistance was detected in 10.1% (41/408) of carbapenem-resistant Enterobacterales. Apologies for the earlier misinterpretation.
Query 9: Looks like there is a Citation errors: Reference [1] is used both for colistin’s resurgence and for WHO’s AMR threat, I suggest to double check the references.
Response: Thank you for the observation and sorry for the mistake. The reference has been corrected accordingly and one reference has been incorporated to correct the error.
Query 10: I think the paper needs major edits for summarizing, revise the whole paper to eliminate the AI terms plus focusing only on the importance of this paper and what question does it answer.
Response: Thank you for your valuable suggestions. I have revised the paper as recommended eliminating AI-related terms, improving summarization, and emphasizing the paper’s significance and research question.
Reviewer 2 Report
Comments and Suggestions for Authors
This work addresses a review that is not abundant in the scientific literature and, in my opinion, is of interest to both researchers and clinicians. In my opinion, the work is comprehensive and organized; however, there are some considerations:
One of the observations that I believe should be incorporated into this review focuses on identifying the differences between the different phosphoethanolamine transferase enzymes. In this regard, although the authors may consider this an important correction, I believe that, since all resistance mechanisms are based on the action of the same enzyme, it would be appropriate to create a figure aligning the sequences of all the enzymes and extracting the percentage of identity between them. This is important so that interested researchers can determine the molecular markers and, consequently, delve deeper into the molecular basis of the mechanisms that confer resistance.
Additionally, the analysis of sequence differences could facilitate the eventual development of treatments with molecules, metabolites, or adjuvant factors. Even with the latest AI tools, the analysis of these differences (in reference to the nucleotide or aminoacids sequences) would facilitate the construction of the structures of these proteins, which could again impact the development of treatments that impact the resistance mechanism by attenuating or nullifying the activity of these proteins, as is currently done using cellular efflux inhibitors to counteract the effect of efflux pumps in antibiotic therapies where bacteria adapt by overexpressing nonspecific mechanisms that, in this example, generally affect cellular homeostasis.
Figure 3 and Table 2 show the same results; it is not possible to include both elements in the work because it results in redundancy.
Figure 4 and Table 4 refer to the same data type; it would be best to incorporate the data from the figure into the table; the map does not contribute anything significant to the work.
The first time these initials are included, it is advisable to include their meaning, for example, Line 341
“To increase colistin resistance testing throughput, accuracy, and speed, improved automated AST”
On line 530
“7.4. Whole Genome Sequencing”
In this regard, it would be interesting to mention cryptic resistance genes, which could be specifically identified (by searching for molecular markers) if the complete genome is available. While it would not be a tool for direct clinical use, it would be the basis for universities to develop a database. This would not only be associated with resistance based on cryptic genes, but also with the pathogenic and virulent potential of the sequenced species. In this regard, I believe this idea, with the addition of some reference, should be included in this section.
Author Response
Reviewer 2:
Query 1: This work addresses a review that is not abundant in the scientific literature and, in my opinion, is of interest to both researchers and clinicians. In my opinion, the work is comprehensive and organized; however, there are some considerations.
Response: Thank you for your valuable feedback. I have addressed your suggestions and revised the manuscript accordingly to enhance clarity and relevance.
Query 2: One of the observations that I believe should be incorporated into this review focuses on identifying the differences between the different phosphoethanolamine transferase enzymes. In this regard, although the authors may consider this an important correction, I believe that, since all resistance mechanisms are based on the action of the same enzyme, it would be appropriate to create a figure aligning the sequences of all the enzymes and extracting the percentage of identity between them. This is important so that interested researchers can determine the molecular markers and, consequently, delve deeper into the molecular basis of the mechanisms that confer resistance.
Response: Thank you for the valuable suggestion. We have now incorporated a phylogenetic tree comparing mcr-1 to mcr-10 to highlight the evolutionary relationships among phosphoethanolamine transferase enzymes. The sequences were aligned, and percentage identities were analyzed to reflect genetic divergence. This figure helps visualize conserved and variable regions among the resistance determinants. These changes have been included in the revised manuscript to support deeper molecular understanding of colistin resistance.
Query 3: Additionally, the analysis of sequence differences could facilitate the eventual development of treatments with molecules, metabolites, or adjuvant factors. Even with the latest AI tools, the analysis of these differences (in reference to the nucleotide or aminoacids sequences) would facilitate the construction of the structures of these proteins, which could again impact the development of treatments that impact the resistance mechanism by attenuating or nullifying the activity of these proteins, as is currently done using cellular efflux inhibitors to counteract the effect of efflux pumps in antibiotic therapies where bacteria adapt by overexpressing nonspecific mechanisms that, in this example, generally affect cellular homeostasis.
Response: Thank you for the insightful comment. I have incorporated the suggestion regarding the relevance of sequence analysis.
Query 4: Figure 3 and Table 2 show the same results; it is not possible to include both elements in the work because it results in redundancy.
Response: Thank you for your observation. I have removed the redundant figure and retained the most informative format to avoid duplication and enhance clarity.
Query 5: Figure 4 and Table 4 refer to the same data type; it would be best to incorporate the data from the figure into the table; the map does not contribute anything significant to the work.
Response: Thank you for the suggestion. I have removed Figure 4 and integrated its data into Table 4 to eliminate redundancy.
Query 6: The first time these initials are included, it is advisable to include their meaning, for example, Line 341 “To increase colistin resistance testing throughput, accuracy, and speed, improved automated AST” On line 530
Response: Thank you for the observation. I have now included the full form “Antimicrobial Susceptibility Testing (AST)” at its first mention to ensure clarity.
Query 7: “7.4. Whole Genome Sequencing” In this regard, it would be interesting to mention cryptic resistance genes, which could be specifically identified (by searching for molecular markers) if the complete genome is available. While it would not be a tool for direct clinical use, it would be the basis for universities to develop a database. This would not only be associated with resistance based on cryptic genes, but also with the pathogenic and virulent potential of the sequenced species. In this regard, I believe this idea, with the addition of some reference, should be included in this section.
Response: Thank you for the valuable suggestion. I have revised Section 7.4 to include the role of WGS in identifying cryptic resistance genes and its potential for database development, along with an appropriate reference.

Reviewer 3 Report
Comments and Suggestions for Authors
After reviewing the manuscript entitled 'Understanding Recent Developments in Colistin Resistance: Mechanisms, Clinical Implications, and Future Perspectives,' a significant question remains: what new insights does this article provide? The mechanism of colistin resistance is well-known, and a simple search in the PubMed database yields 1550 articles describing this mechanism, including the mcr resistance genes. Furthermore, the authors of the manuscript present colistin resistance mediated solely by plasmids, whereas other mechanisms are also involved, such as the overactivation of efflux pumps (https://www.frontiersin.org/journals/microbiology/articles/10.3389/fmicb.2023.1207441/full). Regarding detection methods, there are numerous articles that detail this topic extensively.
Author Response
Reviewer 3:
Query 1: After reviewing the manuscript entitled 'Understanding Recent Developments in Colistin Resistance: Mechanisms, Clinical Implications, and Future Perspectives,' a significant question remains: what new insights does this article provide? The mechanism of colistin resistance is well-known, and a simple search in the PubMed database yields 1550 articles describing this mechanism, including the mcr resistance genes. Furthermore, the authors of the manuscript present colistin resistance mediated solely by plasmids, whereas other mechanisms are also involved, such as the overactivation of efflux pumps (https://www.frontiersin.org/journals/microbiology/articles/10.3389/fmicb.2023.1207441/full). Regarding detection methods, there are numerous articles that detail this topic extensively.
Response: Thank you for your thoughtful comments. In light of your feedback, we have revised the manuscript to emphasize its value in bridging existing knowledge gaps. While mechanisms of colistin resistance are well-documented, our review uniquely integrates these with clinical implications, real-world surveillance limitations, and the underutilization of routine diagnostics in detecting resistance. This revised version aims to serve not just as a summary of current knowledge, but as a strategic and practical resource for guiding future research and clinical efforts in combating colistin resistance.
Round 2
Reviewer 1 Report
Comments and Suggestions for Authors
I noticed multiple instances in the manuscript where the statements do not accurately correspond to the cited references. The manuscript should be thoroughly revised to ensure that each claim is appropriately supported by its respective citation.
Author Response
Reviewer 1:
Query 1: I noticed multiple instances in the manuscript where the statements do not accurately correspond to the cited references. The manuscript should be thoroughly revised to ensure that each claim is appropriately supported by its respective citation.
Response: Thank you for your observation. We have carefully reviewed the manuscript and revised all sections where citations did not accurately support the corresponding statements and highlighted the references which are changed. Appropriate references have been added or corrected to ensure each claim is properly substantiated. We will also rectify any further inaccuracies if identified and thank you for your valuable suggestion.
Reviewer 3 Report
Comments and Suggestions for Authors
Thank you for youre answer
Author Response
Comments1: Thank you for your answer
Reply: Thank you for your kind comments and for the opportunity to revise and improve our manuscript. We sincerely appreciate your time and effort in reviewing our work. We have carefully addressed all comments and made the necessary changes as suggested.